# A Pilot Study on the Microbiome of *Amblyomma hebraeum* Tick Stages Infected and Non-Infected with *Rickettsia africae*

**DOI:** 10.3390/pathogens10080941

**Published:** 2021-07-27

**Authors:** Dalicia Kisten, Jory Brinkerhoff, Selaelo Ivy Tshilwane, Samson Mukaratirwa

**Affiliations:** 1School of Life Sciences, Biological Sciences Section, College of Agriculture, Engineering and Science, University of KwaZulu-Natal, Westville Campus, Durban 4000, South Africa; dalicia3@gmail.com (D.K.); jbrinker@richmond.edu (J.B.); TshilwaneS@ukzn.ac.za (S.I.T.); 2Department of Biology, University of Richmond, Richmond, VA 23173, USA; 3One Health Center for Zoonoses and Tropical Veterinary Medicine, Ross University School of Veterinary Medicine, Basseterre 42123, Saint Kitts and Nevis

**Keywords:** *Amblyomma hebraeum*, bacteria, microbiome, nymph and adult ticks, *Rickettsia africae* infection, *Rickettsia* *aeschlimannii*

## Abstract

Variation in tick microbiota may affect pathogen acquisition and transmission but for many vector species, including *Amblyomma hebraeum*, components and determinants of the microbiome are unidentified. This pilot study aimed to determine baseline microbial community within *A. hebraeum* nymphs infected- and non-infected with *Rickettsia africae* from the environment, and within adult ticks infected- and non-infected with *R. africae* collected from cattle sampled from two locations in the Eastern Cape province of South Africa. Adult A. hebraeum ticks (N = 13) and A. hebraeum nymph (N = 15) preliminary screened for R. africae were randomly selected and subjected to Illumina sequencing targeting the v3–v4 hypervariable regions of the 16S rRNA gene. No significant difference in microbial community composition, as well as rarefied OTU richness and diversity were detected between adults and nymphs. Nymphs showed a higher richness of bacterial taxa indicating blood-feeding could have resulted in loss of microbial diversity during the moulting stage from nymph to adult. Core OTUs that were in at least 50% of nymphs and adults negative and positive for *Rickettsia* at 1% minimum relative abundance were *Rickettsia*, *Coxiella* and *Ruminococcaceae* UCG-005 with a single genus *Arsenophonus* occurring only in nymphs negative for *Rickettsia*. *Ehrlichia* spp. was present in only four nymphal ticks positive for *Rickettsia*. Interestingly, *Rickettsia* *aeschlimannii* was found in one nymph and one adult, indicating the first ever detection of the species in *A. hebraeum*. Furthermore, *A. hebraeum* harboured a *Coxiella*-like endosymbiont, which should be investigated further as *Coxiella* may affect the viability and transmission of other organisms.

## 1. Introduction

Tick-borne diseases are caused by a variety of microbial agents including viruses, protozoa and bacteria [1,2]. Ticks act as reservoirs for a variety of microbial communities and only a few have been studied [3]. Microbial interactions in vectors can affect pathogen transmission or persistence in the vector, hence, the vector microbiome is important to understanding disease risk and dynamics. However, the microbiome is impacted by multiple factors which include host species, location, life stage, and pathogen presence. *Amblyomma hebraeum* is one of the major tick species in South Africa and has been reported to be an efficient vector of a variety of pathogens in animals and humans [4]. It is a three-host-tick belonging to the family Ixodidae [4,5] and apart from transmitting pathogens in livestock and humans it causes major damage to livestock through its bites thereby decreasing animal productivity [4]. The major pathogens transmitted by this tick species are *R. africae* (human) and *Ehrlichia ruminantium* (cattle/wildlife) [4,6]. The genus *Rickettsia* includes numerous tick-borne pathogens [1,7], including some that are zoonotic [8], but it also contains non-pathogenic endosymbiont species [3,9,10]. Although, they are known to be treatable with doxycycline, there remains a difficulty in clinical diagnosis of rickettsiosis, due to the variety of symptoms posed by these organisms [11,12], although, many rickettsioses present with macular erythematous rash that may extend over much of the body surface.

The microbial diversity in a tick may vary in ticks based on factors such as species, geographical location and environmental factors including temperature, humidity, season and habitat type, as well as the stage or sex of the tick and the host(s) involved in the life cycle [13,14]. Although the tick microbiome collectively includes microorganisms, such as viruses, eukaryotes and other tick-borne pathogens, the focal point of microbiome research has been on Eubacteria [9]. Microbiome studies of ticks are increasing in number, and patterns are being identified, but there is still much to be learnt about the factors that affect microbiome composition [15]. 

There is limited information on the microbial communities of these ticks and as to whether certain bacterial families within the microbiome of *A. hebraeum*, may influence the transmission of *Rickettsia* spp. The identification of bacterial communities that constitute the microbiome of a given species of a tick is fundamental to exploring its functions and few studies have been attempted in this regard [2,10]. Studies conducted on *A. hebraeum* have primarily focused on detection of tick-borne pathogens such as *Anaplasma* spp., *Ehrlichia ruminantium*, *Theileria* spp. and *Rickettsia* spp., and hence, there is paucity of research pertaining in the role of the microbiome and the bacteria it constitutes [16]. 

In addition to this, some studies have demonstrated that all stages in the life cycle of *A. hebraeum* are equally able to transmit *Rickettsia* infection to livestock [17]. Due to the current understanding of the microbial community in *A. hebraeum*, it is necessary to understand whether the microbiome differs by life stage of *A. hebraeum* that lead to infection of *R. africae* [17]. Furthermore, it has been argued that although the relative abundance of tick-borne diseases on species has previously been studied as a basis for intervention, the direct and indirect damage from these tick-borne diseases can be avoided with better information on the composition of tick microbial communities and how they affect pathogen transmission [18]. This information can help in understanding the potential transmission of certain species of *Rickettsia,* such as *R. africae* with regard to the locality, seasons and environmental conditions from which they are collected [17,18]. 

This study focused on identifying baseline microbiome information in the nymph and adult stages of *R.*
*africae*-infected and non-*R. africae*-infected *A. hebraeum* from a selected locality in the Eastern Cape province of South Africa. 

## 2. Results

### 2.1. Tick Identification and Status of Infection with R. africae

Some ticks that were preliminarily screened as negative for *Rickettsia* spp. by the process of PCR, tested positive after Illumina sequencing. This could be due to Illumina sequencing being relatively more sensitive compared to PCR, and another reason could be the inability of PCR in identifying certain lineages of *Rickettsia* as the species is so diverse.

Upon morphological classification of ticks sampled from cattle and pastures from Lucingweni, Eastern Cape, no ticks were found that belonged to the genus *Amblyomma*. In total, microbiomes for 28 *A. hebraeum* ticks were characterized: 14 adult *A. hebraeum* were collected from cattle and 14 nymph *A. hebraeum* were collected by drag sampling from Caquba, Eastern Cape (Table A1). PCR and sequencing were carried out for all 28 samples. Analysis of the microbiome data showed a total number of 13,729,994 sequences (range = 10,020–255,352), a mean of 49,036 and median of 22,1715 of which 7850 were unique sequences acquired from all 28 samples. The sequence count showed that even though the sequence variants were different (e.g., different forms of *Rickettsia*) they shared the same genus level. Similar sequences were therefore clustered at 99% similarity to form a final count of 301 OTUs. Sequences that did not fall into any taxon within the Kingdom Bacteria (N = 996) were clustered into an “unassigned” folder.

### 2.2. Microbial Community Structure

The Shapiro-Wilk normality test indicated the data did not follow a normal distribution (*p* = 0.001). Microbiome composition did not vary significantly between nymphs and adults negative for *R. africae* (PerMANOVA: F = 1.0791, *p* = 0.282; Figure 1A) or between nymphs and adults positive for *R. africae* (PerMANOVA: F = 0.7356, *p* = 0.723; Figure 1B) as well as between all nymphs (positive and negative) and adults (positive and negative) (PerMANOVA: F = 0.8795, *p* = 0.492; Figure 1C). Rarefied OTU richness did not vary significantly between *A. hebraeum* nymphs and adults, which were negative for *R. africae* (*p* > 0.05, T = −1.70; Figure 2), as well as between nymphs and adults which were positive for *R. africae* (*p* > 0.05, T = −0.93; Figure 2). 

The analysis of alpha diversity measures using a pairwise-Wilcoxon test showed no significant difference between negative nymphs and adults (*p* > 0.05, T = 1.02) (Figure A1 in Appendix A) and positive nymphs and adults (*p* > 0.05, T = 0.42). Using raw data, the anosim R value when comparing the mean of ranked dissimilarities within groups to the mean of ranked dissimilarities between groups indicated an even distribution of high and low ranks for both negative and positive nymphs, and adults, respectively (R statistic = −0.04; R statistic = 0.36). The closer the R value is to zero the more similar the microbial composition between adults and nymphs. Negative nymphs and adults showed no significant difference in microbial communities between and within groups (*p* = 0.58). While positive nymphs and adults showed statistically significant difference in microbial communities within groups rather than between (*p* = 0.02). The complemented NMDS plot (Figure 3) showed an overlap between OTU bacterial composition for negative nymphs and adults and positive nymphs and adults which correlated with anoism and PerMANOVA outputs. The stress of 0.104 indicated a good level of ordination in terms of reduced dimensions. Rarefaction curves (Figure A2A,B) are illustrated using the mean species diversity of bacterial composition in OTUs for positive nymphs and adults, and negative nymphs and adults. Both rarefaction curves indicated that majority of samples did not approach the saturation plateau, and hence, there was not enough coverage to represent all bacterial populations.

### 2.3. Taxonomic Differences

The initial summary of OTUs for the negative group was 301 taxonomic classifications in 15 samples and for positive group, 301 taxonomic classifications in 13 samples. After utilizing the phyloseq prune function the taxa were reduced to 271 taxonomic classifications in the negative group and 226 taxonomic classifications in the positive group. Phyla that showed the highest relative abundance within the negative and positive groups respectively were Firmicutes (33.95%) and (35.40%), Proteobacteria (29.52%) and (30.09%), Bacteroidetes (17.34%) and (16.81%) followed by lower percentages in Actinobacteria (7.01%) and (5.75) and Tenericutes (2.58%) and (3.10%) and some phyla contributed percentages of 1% and below (Figure 4). The genus *Rickettsia* (phylum Proteobacteria) was detected in 96.43% of ticks (27 of 28) across all samples including samples that initially tested negative during PCR (Figure 5). Nymphs had a higher richness of microbial communities as compared to adults (Figure 6). Further analysis of relative richness of bacterial community boxplots using the phyloseq psmelt function separated the adults from the nymphs for negative and positive samples (Figure A3) emphasizing the high counts of Firmicutes, Proteobacteria and Bacteroidetes. A Venn diagram was used to evaluate the similarities between the two groups, defined as core OTUs at genus level present in at least 50% of negative nymphs and adults (Figure 7A) and positive nymphs and adults (Figure 7B) at 1% minimum relative abundance. Only one genus, *Arsenophonus*, belonging to the phylum Proteobacteria was specific to negative nymphs whilst the overlap revealed that both nymphs and adults shared genera *Coxiella* (Proteobacteria), Ruminococcaceae UCG-005 (Firmicutes), *Rickettsia* (Proteobacteria) and “Other” (Sequences that were not classified into a known phylum). There were no genera distinctive to either positive nymphs or adults, whilst the overlap showed they shared genera *Rickettsia*, *Coxiella*, family Ruminococcaceae UCG-005 and “Other”. 

## 3. Discussion

The aim of this study was to determine the microbiota community present within *A. hebraeum* nymphs (collected from the environment), and adult ticks (collected from cattle) infected with *R. africae* and from the non-infected stages in Caquba locality in the Eastern Cape province of South Africa. The results from the study showed no significant differences in OTU richness and diversity between nymphs and adults positive and negative for *R. africae*. This was corroborated by the minimal sequencing depth of the samples with most OTUs having a bacterial composition less than 50,000 sequences and only a few OTUs having more than 200,000 sequences. As previously demonstrated, a compromise must be made between the precise measure of species diversity and the relative abundance and how rarefaction is used to lessen the negative outcome of uneven sampling effort [19]. A previous study revealed similar results to our study regarding lack of OTU richness and diversity of *Anaplasma*/*Ehrlichia* positive and negative ticks although there was presence of variation in represented OTUs [20].

Some samples that were initially screened as negative for *R. africae* during PCR [21] were later shown to be positive during Illumina sequencing probably due to the process being relatively more sensitive and able to analyze trace amounts of *Rickettsia* sequences and diverse lineages [22]. It has been highlighted in some studies that commercially supplied DNA extraction kits used for PCR could possibly contain bacteria, which negatively hinder the downstream analysis during sequencing hence this may have affected the *Rickettsia* status as well [23]. 

Our study showed that microbial constituents of nymphs and adult ticks were shared between the two life stages and offered very little variation. Insignificant difference was found between microbiome variation between nymphs and adults. This is likely due to the close geographic proximity of the sampling sites in Caquba as microbial diversity in ticks has been shown to vary with geographic location, environmental factors and species identity. Various studies have shown that the closer sampling sites are to each other, the less distinct the microbial composition within a particular tick species is likely to be [13,14,19]. 

Tick microorganism variation is likely to be increased by those organisms transmitted from the animal to the tick whilst blood feeding, as well as the soil or pasture that immature stages are sampled from [3]. This is supported by results [20], which showed that tick microbiota differed at sites with different soil types, but soil type was not considered in this study, as well as the species of tick host, including sampling of different regions. 

Analysis of OTU relative abundance showed a total of fourteen phyla that were shared between nymphs and adults. From the fourteen phyla, Firmicutes, Proteobacteria and Bacteroidetes dominated the samples in this study, and this agrees with previous studies, which showed that tick microbiota found in *A. hebraeum* mainly consists of the same three phyla in varying amounts [9,13,14,24]. Similar results have also been reported regarding the microbiome of *A. americanum* [19,20] and of *Amblyomma maculatum* [25]. 

Positive nymphs and adults had a higher composition of Firmicutes compared to negative nymphs and adults. Within the phylum Firmicutes were 44 genera and 17 species with one of the core genera being Ruminococcaceae UCG-005 with an unclassified species. Nymphs had a high relative abundance of Orders Bacilalles and Clostridiales and in accordance with a previous study by Clow et al. [26] who reported Ruminococcaceae as one of the most dominant families in Ixodes scapularis and Dermacentor variabilis. Clostridiales has also been reported as the dominating group in tick species infesting livestock in Pakistan [27]. 

Proteobacteria accounted for the second highest percentage of bacterial composition with the genus *Rickettsia* having the highest abundance. They were detected in 96.43% of ticks (27 of 28) across all samples including samples that initially tested negative during PCR. It was assumed *Rickettsia* spp. would dominate the samples as *A. hebraeum* is the principal vector of *R. africae*, evidently the study conducted by Magaia et al. [28] confirmed that 80% of *A. hebraeum* ticks were infected with *R. africae* in Mozambique. As with other taxa presented in this study, nymphs had a higher relative abundance of *Rickettsia* spp. and this is consistent with report by Jongejan et al. [18] who found that *R. africae* was present in adult and nymph *Amblyomma* ticks at 15.7%, and 40%, respectively. The high relative abundance of *Rickettsia* spp. in nymphs can also be explained by the vertical transmission of the bacterium from egg masses to larvae then to nymph [7,19,29]. By comparing the life stages, nymphs showed richness in bacterial composition when compared to adults and this could be due to the possibility of the loss of some bacterial composition from the time the nymph starts to moult to 100 days after it has reached its adult stage [19,30].

We anticipated a higher percentage of *Ehrlichia* spp. positive samples in this study, however, the family Anaplasmataceae were present in only four nymphs. This is in accordance with Jongejan et al. [18] who reported co-infection of *E. ruminantium* and *R. africae* in 14.9% of nymphs in South Africa. The finding of *Ehrlichia* spp. in nymphs collected from the environment is in contrast with Mtshali et al. [31]. who found that questing nymphs were negative for *Ehrlichia* spp. 

Interestingly, *R. aeschlimannii* was found in a single *A. hebraeum* adult and nymph. There are no previous reports that have indicated *A. hebraeum* being a carrier for *R. aeschlimannii*. A study conducted by Pretorius et al. [32] reported the first occurrence of *R. aeschlimannii* in *R. appendiculatus* in South Africa. *Rhipicephalus* (*B*.) *microplus* and an unidentified *Rhipicephalus* species were present in cattle along with *A. hebraeum.* Whilst, sampling in our study area and a possible transference of microbiota between two different species of ticks during blood feeding on the same host is uncertain but not unlikely. Comparably, significant microbiome variation in *I. scapularis* and *D. variabilis* has been reported even though the ticks fed on the same host [24]. Further analysis on the genus *Rhipicephalus* sampled from the same location could provide insight on the *Rickettsia* lineages and the presence of *R. aeschlimannii*. 

A singular genus specific to only nymphs negative for *Rickettsia* at 50% relative abundance was *Arsenophonus* belonging to the phylum Proteobacteria. Notably, *Arsenophonus nasoniae* has been associated with ticks belonging to the genera *Amblyomma* and *Dermacentor* locate in the United States of America [10]. The extensive distribution of *Arsenophonus*-like endosymbionts within *A. americanum* was similar to that found within *D. variabilis* and molecular analysis of Ixodes ricinus ticks, showed 37% of nymphs contained a strain of the *Arsenophonus* bacterium as compared to the adult ticks which contained only 3.6% [33]. The study proposed that the occurrence of *Arsenophonus* is much higher in the immature stages of these hard ticks, which might explain its presence in the nymphal stages in the present study. 

The genus *Coxiella* also belonging the phylum Proteobacteria had Jongejan et al. [18] infection rate in all ticks (nymph and adult). This outcome is comparable with Mtshali et al. [31] who found absence of *C. burnetti* in ticks sampled from Eastern Cape Province of South Africa but was present in ticks sampled from Kwa-Zulu Natal, Free State and Mpumalanga provinces of South Africa. Although, the infection rate recorded was 7%. The species of the *Coxiella* symbiont in the present study could not be identified however its suggestive it could be closely related to the most common species in South Africa being *C. burnetti* the causative agent for Q fever, which has been commonly detected in numerous hard ticks one of them being the genus *Amblyomma* [10,34]. Evidence provided by Simpson et al. [34] indicated a high seroprevalence of 60.9% for Q fever in farmers and 37.8% in malaria-negative acute febrile illness patients in Mpumalanga. The genus *Coxiella* has been commonly found in *A. hebraeum*, prevalent across many locations across South Africa, vertically transmitted and is considered an endosymbiont of potential importance [10,20,34]. 

It is important to note that non-pathogenic organisms play an important role within the tick microbiome and specific non-pathogenic symbionts affect tick-borne pathogen transmission through competition or gene transfer [20]. For instance, *Coxiella* and *Rickettsia* spp. are considered non-pathogenic, but some of them are known to be tick-borne pathogens which affect humans and animals [20]. *Coxiella* spp. identified in their study by Trout-Fryxell and DeBruyn [20] may be an obligate endosymbiont. Although, it was only present in 74% of samples, this inherently suggested that if *Coxiella* is dependent on the tick host, the tick host is not dependent on *Coxiella*. Similarly in this study, *Coxiella* was present in 96.43% of specimens meaning it is not an obligate endosymbiont.

Authors should discuss the results and how they can be interpreted from the perspective of previous studies and of the working hypotheses. The findings and their implications should be discussed in the broadest context possible. Future research directions may also be highlighted.

## 4. Materials and Methods

### 4.1. Study Site

Sampling of *A. hebraeum* ticks was conducted between July 2018 to July 2019 in the Eastern Cape Province of South Africa. Ticks were collected from two study sites namely, Lucingweni (latitude/longitude 31.459111/28.756333) in Mthatha which is inland, and Caquba (latitude/longitude 31.6425/29.460028) in Port St Johns located along the coastline as previously described [21]. The sampling entailed tick collection from cattle (partly engorged adult ticks) and nymphs from the environment at 2 replicate quadrants at each site where quadrant was 50 × 100 m. 

### 4.2. Tick Collection

Ticks questing on pastures were collected by drag sampling which entailed dragging 2.5 m × 1.5 m cloths through the vegetation along a 50 × 100 m quadrant. Nymphs were removed from the drag cloth using forceps and transferred into a glass vial containing 70 % ethanol for further processing. 

At each location, indigenous cattle were herded into a cattle crush/dipping race and once secured, adult *A. hebraeum* ticks were physically pulled-off the cattle using blunt-tipped forceps. Common sites of predilection of attachment of the ticks were the legs, tail, upper perineum (base of the tail and below the anus) and lower perineum (base of the scrotum). Ticks collected from each animal were placed in separate labelled glass vials containing 70% ethanol. On the label, we recorded the identification number of the animal, the sample site, date of collection and name of collector. The vials with preserved ticks were kept at room temperature until they were processed in the laboratory. 

### 4.3. Processing of Ticks for Analysis

Collected ticks were morphologically identified to species level [5]. The samples were then frozen at −80 °C until DNA extraction. Before DNA extraction, samples were removed from the freezer and allowed to thaw at room temperature. An amount of 600 µL of 95% EtOH was added to the sample tubes and vortexed for 30 seconds. The EtOH was decanted, and the tube was refilled with 600 µL of distilled water and vortexed for 15 s. The distilled water was decanted, and the samples were air-dried for 10 min. The ticks were bisected from the mouth parts to the festoons. One half of the tick was placed into a 1.5 mL Eppendorf tube for DNA extraction and the other half stored in the freezer at 4 °C till further use. Glass beads of 0.1 mm diameter were placed in Eppendorf tubes containing tick samples for DNA extraction and then placed in a bead beater (Disruptor Genie, Scientific Industries, Inc, New York, NY, USA) for 2 min at 3000 rpm.

### 4.4. DNA Extraction

Half the portion of each tick was screened for *R. africae* infection using an established protocol and published primers [21] and from the outcome of screening (Table A1), 13 positive (8 adults and 5 nymphs) and 15 negative nymphs and adult (6 adults and 9 nymph) *A. hebraeum* were randomly selected from the screened pool for DNA extraction before running PCR with 16S rRNA primers. DNA extraction was done using ZYMO Quick-DNA Miniprep Plus Kit (ZYMO Research, Orange, CA, USA) according to a modified version of the manufacturer’s protocol. Samples were incubated at 56 °C for 12 h until the sample tissue was fully digested instead of three hours as stated in the protocol. 

### 4.5. PCR Amplification of Bacterial Taxa in A. hebraeum Ticks

A 16S rRNA gene sequence analysis for the detection of sequence differences in the hypervariable regions, which are present in all bacteria, was used. Studies show that hypervariable V3–V4 regions show the most representative taxonomic bacterial profiles found in ticks [35]. PCR conditions were carried out following the protocol described in 16S Metagenomic Sequencing Library Preparation manual (Illumina, 2013). Locus-specific primers were attached to Illumina overhang adapter nucleotide sequences; forward primer.

5′–TCGTCGGCAGCGTCAGATGTGTATAAGAGACAGCCTACGGGNGGCWGCAG–3′ and reverse primer: 5′–GTCTCGTGGGCTCGGAGATGTGTATAAG AGACAGGACTACHVGGGTATCTAATCC –3′ (Illumina, 2013). The 25 μL reaction mixture contained 2.5 μL sample DNA, 12.5 μL Taq Polymerase (0.5 U), 5 μL forward primer (1 μM) and 5 μL reverse primer (1 μM). For control sample: DNA was replaced with nuclease free water. Samples were vortexed for 30 s. Samples were then placed in the thermocycler S1000 Thermal Cycler (Bio-Rad, Hercules, CA, USA) for PCR according to the following conditions: 1 cycle initial denaturation for 3 min at 95 °C, 25 cycles denaturation for 30 seconds at 95 °C, annealing for 30 seconds at 55 °C, extension for 30 seconds at 72 °C and final extension for 5 minutes at 72 °C (Illumina, 2013). The PCR product together with 1kb DNA ladder molecular marker (GeneRuler DNA Ladders, Thermo Fisher Scientific Inc., MA, USA) were run in electrophoresis using a 1% agarose gel at 80V for 60 minutes (Powerpac Basic Power Supply, Bio-Rad, Hercules, CA, USA). The band size for the 16S rRNA gene, including adapters and index sequences was approximately 500 base pairs. The PCR amplicons were sent to National Health Laboratory Service for PCR product purification, quantification and Illumina sequencing. 

### 4.6. Informatic and Statistical Analysis

The bioinformatic analysis was conducted in Quantitative Insights into Microbial Ecology 2 ($\mathrm{QIIME}\;2^{\mathrm{TM}}$) [36]. Forward and reverse raw fastq sequence reads were first merged into individual contigs. The paired reads were subjected to a series of quality control steps where low-quality reads, as well as reads that did not meet the minimum length criterion of 440 base-pairs were filtered out and singletons were removed. Operational taxonomic units (OTUs) were clustered using QIIME 2. The OTUs were assigned using closed reference picking and taxa was assigned using the Silva reference database [37] (Quast et al., 2012). The program USEARCH v.11 [38] was used to remove chimeras and cluster sequences that showed 99% similarity into the same OTU. 

Statistical analysis was conducted on software R v.4.0.2 [39]. Alpha and beta diversity measures, analysing variation in bacterial diversity (Shannon index) and richness were carried out using the Wilcoxon test for pairwise comparisons. Statistical significance was determined by comparison between negative nymphs and adults, as well as positive nymphs and adults under the null hypothesis of no significant difference between groups. The vegan package v.4.5.6 [40] was used to analyse diversity of bacterial communities by performing permutational multivariate analysis of variance (PerMANOVA, adonis function, permutations = 999, set seed = 1). Weighted unifraction plots were produced by comparing microbial composition in *A. hebraeum* nymphs and adults that tested positive for *Rickettsia*, nymphs and adults that tested negative for *Rickettsia* and a plot with combined (positive and negative) nymphs and adults. An anosim test was performed to test for dissimilarity of microbial communities between and within groups. A Non-metric Multi-dimensional Scaling (NMDS) plot was generated using the vegan package v.4.5.6 [40] to visually support PerMANOVA and anosim outputs. Variable sequencing depth was accounted for by the rarefying raw OTU counts for nymphs and adults negative for *Rickettsia* and nymphs and adults positive for *Rickettsia*. All sequences were classified from phylum to genus level using R phyloseq package [41]. The relative abundance bar plots representing phyla from negative and positive nymphs and adults were generated using the Phyloseq R package. Individual relative abundance phyla boxplots were generated using the phyloseq psmelt function. Core microbiota was evaluated at 50% presence of taxa in the negative and positive groups at 1% minimum relative abundance, this was accomplished using the R microbiome package v1.10 [42].

### 4.7. Ethical Statement

The study was approved by the University of KwaZulu-Natal Animal Research Ethics Committee, as well as the Department of Agriculture, Land Reform and Rural Development (DALRRD); previously Department of Agriculture, Forestry and Fisheries (DAFF) under Section 20 of the Animal Diseases Act of 1984 (Act No 35 of 1984).

## 5. Conclusions

The evidence presented in this study suggested that *A. hebraeum* nymphs sampled from Caquba in the Eastern Cape province of South Africa sustained a multifaceted microbiome, which was mainly composed of non-pathogenic microorganisms, and some which can cause tick-borne diseases. The study indicates that *A. hebraeum* is a definitive reservoir and vector for *Rickettsia* spp. in the Eastern Cape province of South Africa and should therefore be regarded as of zoonotic importance. It is evident that *A. hebraeum* harboured a *Coxiella*-like endosymbiont which should be researched further as *Coxiella* may affect the viability and transmission of other organisms. Multiple infections with these pathogens may result in increased bacterial burden, as well as more severe and diverse clinical manifestation of disease in livestock. A difficulty arises in the analysis and identification of microbial constituents in *A. hebraeum* due to the excessive abundance of *Rickettsia* and *Coxiella* spp. but this can be improved with an increase in sampling effort and thereby sequencing depth. Evidence regarding bacteria which may inhibit or promote the transmission of *Rickettsia* remains inconclusive. However, vector control strategies should be considered. The presence of *R*. *aeschlimannii* is the first ever to be recorded in *A. hebraeum* and further studies should be based on a comparison between the microbiome variation in *Rhipicephalus* spp. and *A. hebraeum*. 

## Figures and Tables

**Figure 1 pathogens-10-00941-f001:**
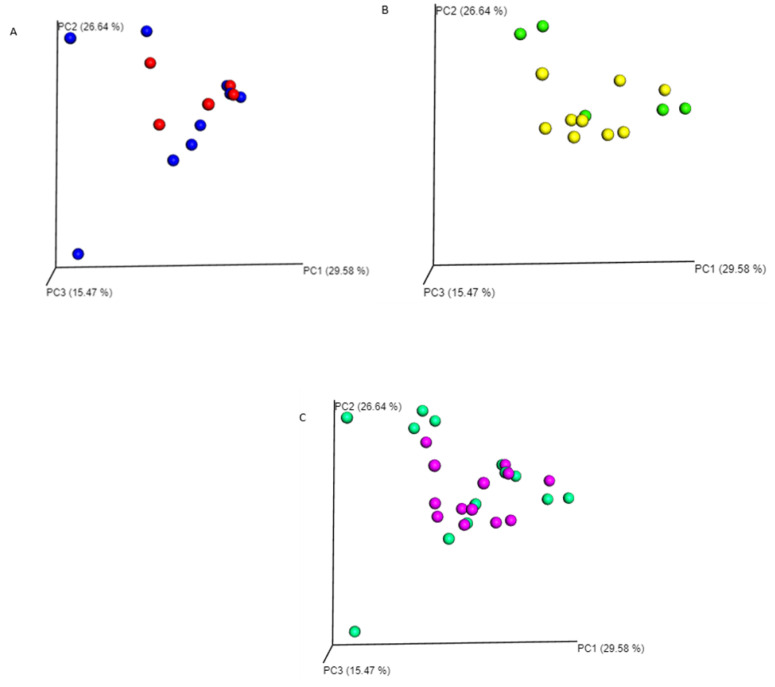
Weighted unifraction plots for *Amblyomma hebraeum* nymphs and adults negative and positive for *Rickettsia africae*. (**A**)-Negative nymphs and adults (adults are red, nymphs are blue), (**B**)-positive nymphs and adults (adults are yellow, nymphs are green), (**C**)-Combined nymphs and adults (positive and negative) (adults are purple, nymphs are green).

**Figure 2 pathogens-10-00941-f002:**
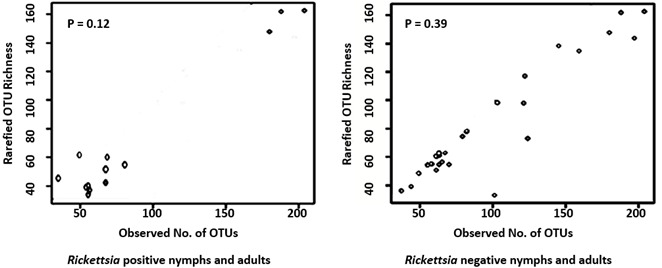
Rarefied OTU richness of *Amblyomma hebraeum* nymphs and adults negative and positive for *Rickettsia* africae. Dots indicate observed OTUs. The *p*-value indicates the significance level.

**Figure 3 pathogens-10-00941-f003:**
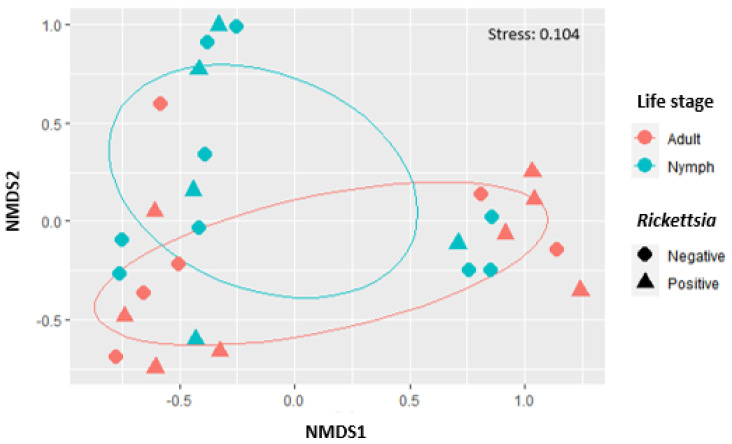
Non-metric multidimensional scaling (NMDS) ordination plot of OTU bacterial composition in *Amblyomma hebraeum* nymphs and adults negative and positive for *Rickettsia* africae. Colors pink and blue indicate life stage and shapes indicate *Rickettsia* status. The stress value was 0.104. Ellipses indicate the 95% confidence interval around the centroid in non-dimensional space.

**Figure 4 pathogens-10-00941-f004:**
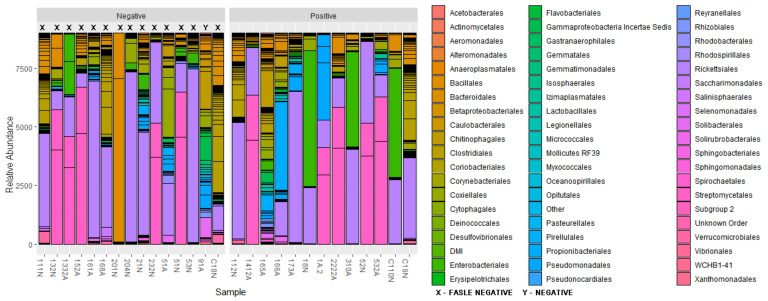
Relative abundance of bacterial communities according to order classification for *Amblyomma hebraeum* nymphs and adults negative and positive for *Rickettsia africae*. X–False negative samples, Y–Negative sample.

**Figure 5 pathogens-10-00941-f005:**
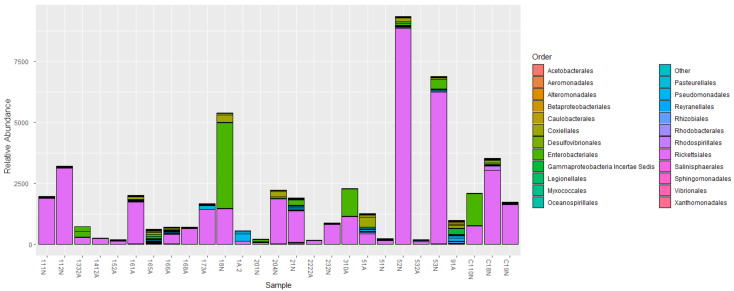
Relative abundance of bacterial communities according to order classification within the phylum Proteobacteria for *Amblyomma hebraeum* nymphs and adults negative and positive for *Rickettsia africae.*

**Figure 6 pathogens-10-00941-f006:**
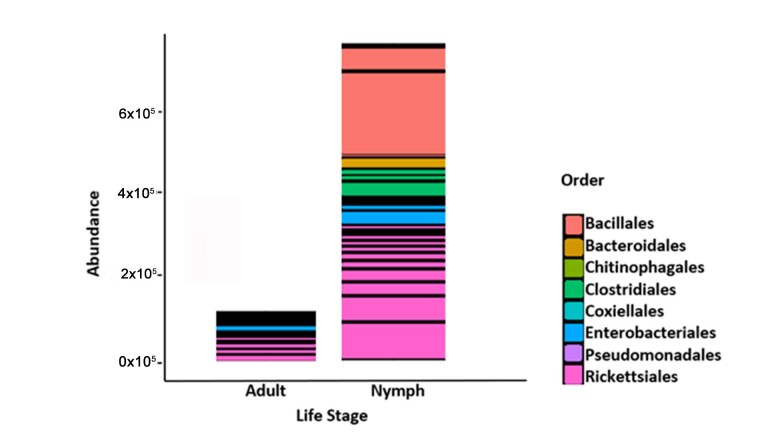
Relative abundance bar plot representing bacterial communities according to order classification for *Amblyomma hebraeum* nymphs and adults negative and positive for *Rickettsia africae*.

**Figure 7 pathogens-10-00941-f007:**
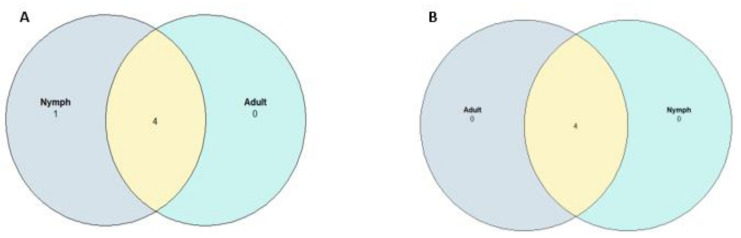
Core OTUs defined as genus present in at least 50% of *Amblyomma hebraeum* nymphs and adults negative and positive for *Rickettsia africae* at 1% minimum relative abundance. (**A**) Nymphs and adults negative for *Rickettsia,* one genus (*Arsenophonus*) specific to negative nymphs (A-overlap)-*Coxiella*, Ruminococcaceae UCG-005, *Rickettsia* and “Other”. (**B**) Nymphs and adults positive for *Rickettsia* (B-overlap)-*Coxiella*, Ruminococcaceae UCG-005, Rickettsia and “Other”.

## Data Availability

The data presented in this study are contained within the article.

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
