# Peer review of "A Pilot Study on the Microbiome of Amblyomma hebraeum Tick Stages Infected and Non-Infected with Rickettsia africae"

_pathogens, 2021, doi:10.3390/pathogens10080941_

Round 1

Reviewer 1 Report

The manuscript reports a study to establish the possibly changes of microbiome composition of Amblyomma hebraeum ticks developmental stages in association to Rickettsia africae infection. In my opinion it is an interesting and original manuscript. In general article is well written. The data are certainly valuable and will add to the knowledge of scientific community. The research design and execution is of good standard. The analysed problems are clearly presented. All methods are correctly used.

I have an objection to the sample size. 28 ticks from two research plot is definitely not enough to draw general conclusions and obtain statistically significant results. Despite the high quality of the techniques used and the perfectly written text, the manuscript can be rejected on the base of the objection. I suggest you study a much larger sample, a minimum of 50 nymphs and adults from single plot. An alternative is to shorten the manuscript and present the current results in the form of case reports.

I have some specific comments (see below):

12 – hebreum change to hebraeum

40 – “more than 650 species”  - Guglielmone et al. 2014 (The Hard Ticks of the World. Springer) estimate over 700 hard ticks species; Estrada-Pena et al. 2017 (Ticks of Europe and NorthAfrica, Springer) estimate about 660 hard ticks species. Maybe ignore this information at all? This information is not needed to correctly interpret the results of the described tests.

101 – Rickettsia spp. - generic names should be in italics. The same error is repeated in the rest of the text (107, 108, 121 etc.)

239 – study by [26] – should be study by Clow et al. [26]. The same error is repeated in the rest of the text (260, 264 etc.)

311-315. Study site – currently, it is standard to provide geographic coordinates - length and width.

Author Response

We have included our point-by-point responses to the Reviewer on the attached file.

Reviewer 2 Report

General Comments

The present work aimed to investigate the bacterial community within A. hebraeum nymphs infected- and non-infected with R. africae from the environment, and within adult ticks infected- and non-infected with R. africae collected from cattle sampled in South Africa. For this purpose, authors sequences the V3-V4 hypervariable region of 16S rRNA.

Specific Comments

  • Authors should submit the article to a English revision, mainly regarding the use of comma.
  • Scientific names and genera should be italicized throughout the MS.
  • Line 12: hebraeum
  • Were the adult ticks collected from cattle totally or partly engorged?
  • What gene was used for screening for R. africae? Is this PCR specific for R. africae? Were the amplicons sequenced?
  • Line 115: Authors are saying here that they are not sure the Rickettsia species detected in ticks were R. africae? Confusing…
  • Did the authors sequence the positive samples for Rickettsia? Those samples negative in PCR BUT positive in deep sequencing (Illumina) were grouped into Rickettsia positive samples? This is not clear… how many sequences were in total positive for Rickettsia (by PCR or Illumina)? Even though authors detected Rickettsia (by Illumina) in previously Rickettsia-PCR negative ticks, those ticks were grouped in R. africae-negative samples? 
  • Did the authors compare the microbiome composition of positive x negative nymphs and positive x negative adults? This is not clear in the MS.
  • Line 161: If some samples (how many?) were positive for Rickettsia after deep sequencing, how come the number of positive samples is lower than that of negative samples?
  • Authors mentioned in Line 231 that all nymphs and adults were positive for Coxiella, but in Line 304 they said they found Coxiella in 96.43 of the evaluated specimens???
    1. Lines 305-308: ???
  • Line 347: Which modifications?
  • Lines 360-361: please, indicate the concentration of used primers (instead of volume) and how many U of Taq were used
  • Line 422: Rhipicephalus spp.

Author Response

We have provided point-by-point Response to Reviewer's comments/suggestions and we are grateful for the constructive comments.

Reviewer 3 Report

To Authors,

This paper shows the relation between tick microbiome and Rickettsia. The result did not support your hypothesis showing different microbiome between Rickettsia-positive ticks and Rickettsia-negative ticks, but this data would be valuable.

Major comments

In the present paper, Introduction is redundant. Some sentences showing similar meanings were observed in different paragraphs. Especially in L. 58-84, there are many overlapping to first paragraph and between the paragraphs.

You often use the terms of “positive” and “negative” as a meaning “Rickettsia positive” and “Rickettsia negative” in Result section (after L. 125). Describe sentences with accuracy.

  1. 164-167. You describe that the highest relative abundance of phyla is Firmicutes. In Fig. 4, however, the bar would show that the highest abundance is Proteobacteria. What was the cause of this consistency?

In Fig. 4, how about the relative abundance of Rickettsia in the proteobacteria phylum? When you remove the reads of rickettsia, the pattern of diagram may change in Rickettsia positive ticks.

L.208-214. You describe that you diagnose some ticks as Rickettsia-negative in the first screening but illumine sequencing shows some of Rickettsia-negative was Rickettsia-positive. Which sample is the false negative in figures (such as Fig. 1 and 4)? When you trust illumine sequencing, the results did not change? I think that you should show the which plot in Fig.1 and which bar in Fig.4 was false negative.

Was the raw data of amplicon sequencing deposited?

Minor comments

  1. 14. R. africae -> Rickettia africae
  2. 125. A. hebraeum -> R. africae

In Figure 2, describe legends of horizontal axis. Which plot show nymph or adult?

Author Response

We have provided a point-by-point response to Reviewer's comments/suggestions and we are grateful to the constructive suggestions.

Round 2

Reviewer 2 Report

Authors addressed all the comments raised by this reviewer.

Reviewer 3 Report

The authors have made substantial revisions to the manuscript and have addressed most of comments that I had previous version. And, the title including “a pilot study” is very good idea. I agree with the other reviewers accepted.